# Metacarpophalangeal Joint Pathology and Bone Mineral Density Increase with Exercise but Not with Incidence of Proximal Sesamoid Bone Fracture in Thoroughbred Racehorses

**DOI:** 10.3390/ani13050827

**Published:** 2023-02-24

**Authors:** Kira J. Noordwijk, Leyi Chen, Bianca D. Ruspi, Sydney Schurer, Brittany Papa, Diana C. Fasanello, Sean P. McDonough, Scott E. Palmer, Ian R. Porter, Parminder S. Basran, Eve Donnelly, Heidi L. Reesink

**Affiliations:** 1Department of Clinical Sciences, College of Veterinary Medicine, Cornell University, Ithaca, NY 14853, USA; 2Department of Materials Science and Engineering, College of Engineering, Cornell University, Ithaca, NY 14853, USA; 3Department of Biomedical Sciences, College of Veterinary Medicine, Cornell University, Ithaca, NY 14853, USA; 4Department of Population Medicine and Diagnostic Sciences, College of Veterinary Medicine, Cornell University, Ithaca, NY 14853, USA

**Keywords:** ash fraction, computed tomography, dual-energy X-ray absorptiometry, Raman spectroscopy, bone mineral density

## Abstract

**Simple Summary:**

Identifying imaging features associated with racehorse catastrophic musculoskeletal injuries could improve both jockey and racehorse welfare. Bone mineralization and fetlock joint pathology have been hypothesized to correlate with catastrophic proximal sesamoid bone (PSB) fracture. This study compared bone mineral properties in cadaver distal limb specimens obtained from horses sustaining catastrophic PSB fracture and controls using multiple imaging modalities, including computed tomography (CT), dual-energy X-ray absorptiometry (DXA), and Raman spectroscopy. Fetlock joint pathology was compared between fracture and control groups using CT. Few differences were observed between PSB fracture and control groups; however, total high-speed furlong exercise was strongly predictive of third metacarpal bone mineral density and pathologic features, including palmar osteochondral disease (POD), condylar sclerosis, and condylar lysis. Total high-speed furlong exercise was also predictive of increased radiographic bone density in the subchondral region of PSBs.

**Abstract:**

Proximal sesamoid bone (PSB) fracture is the leading cause of fatal musculoskeletal injury in Thoroughbred racehorses in Hong Kong and the US. Efforts are underway to investigate diagnostic modalities that could help identify racehorses at increased risk of fracture; however, features associated with PSB fracture risk are still poorly understood. The objectives of this study were to (1) investigate third metacarpal (MC3) and PSB density and mineral content using dual-energy X-ray absorptiometry (DXA), computed tomography (CT), Raman spectroscopy, and ash fraction measurements, and (2) investigate PSB quality and metacarpophalangeal joint (MCPJ) pathology using Raman spectroscopy and CT. Forelimbs were collected from 29 Thoroughbred racehorse cadavers (*n* = 14 PSB fracture, *n* = 15 control) for DXA and CT imaging, and PSBs were sectioned for Raman spectroscopy and ash fraction measurements. Bone mineral density (BMD) was greater in MC3 condyles and PSBs of horses with more high-speed furlongs. MCPJ pathology, including palmar osteochondral disease (POD), MC3 condylar sclerosis, and MC3 subchondral lysis were greater in horses with more high-speed furlongs. There were no differences in BMD or Raman parameters between fracture and control groups; however, Raman spectroscopy and ash fraction measurements revealed regional differences in PSB BMD and tissue composition. Many parameters, including MC3 and PSB bone mineral density, were strongly correlated with total high-speed furlongs.

## 1. Introduction

The metacarpophalangeal joint (MCPJ) is the most common site for fracture in Thoroughbred racehorses [1,2], and proximal sesamoid bone (PSB) fracture is the leading cause of fatal musculoskeletal injury in Hong Kong and the US [1,2,3,4]. Prior studies have evaluated exercise history and PSB morphology [5,6,7,8,9] using radiography [7,9,10] and computed tomography (CT) [9,11,12,13,14] to identify risk factors and imaging features associated with PSB fracture. Bone volume fraction [11,14], bone mineral density [8,9,12], sclerosis [9,15], and focal osteopenic lesions [8,9] have been investigated as potential imaging features associated with catastrophic PSB fracture [8,9,11,12,14,15]. Here, we investigated multiple modalities for assessing metacarpophalangeal joint pathology and bone mineralization in Thoroughbred racehorses sustaining catastrophic PSB fracture and controls, including dual-energy X-ray absorptiometry (DXA), computed tomography (CT), Raman spectroscopy, and ash fraction measurements.

In humans, bone mineral density (BMD) is the most important quantitative predictor of fracture risk [16]. UK racehorses sustaining lateral condylar fracture of the third metacarpal bone (MC3) had increased MC3 BMD adjacent to the fracture site as compared to the non-fractured condyle or non-fracture racehorse control MC3 [17]. Conversely, in New Zealand racehorses, MC3 BMD did not differ between horses with condylar fractures and non-fracture controls but was greater in horses with more exercise [18,19]. In New York racehorses, microCT-derived PSB bone volume fraction (BVF) was greater in horses sustaining catastrophic PSB fracture as compared to controls [11]. Similarly, voxel-based morphometry revealed increased PSB BMD in Hong Kong racehorses with catastrophic PSB fracture as compared to controls [12]. In California racehorses, a focal subchondral region of osteopenia and decreased tissue mineral density was identified in the abaxial, subchondral region of medial PSBs from racehorses that sustained catastrophic PSB fracture which was hypothesized to predispose to PSB fracture [9].

Imaging modalities capable of quantifying bone mineral content in vivo include dual energy X-ray absorptiometry (DXA) and quantitative computed tomography (CT). DXA is the most common osteoporosis screening method in women [20,21], and DXA has demonstrated comparable accuracy to ash fraction measurements for quantifying bone mineral density of the mid-MC3 in an equine cadaver study [22]. Likewise, a portable DXA device was used to measure BMD in the live, standing horse, suggesting that DXA could be adapted to equine practice for in vivo detection of BMD changes [22]. Raman spectroscopy is being investigated for clinical use to predict age-related fragility fractures in humans [23,24] and has been used to evaluate the equine MC3 in cadaveric tissues [25] as well as in vivo [26].

Radiography has been used to evaluate equine MCPJ pathology, including changes associated with palmar osteochondral disease (POD) [27] and fetlock fractures [28]. Prior studies have identified gross [6,9,11], radiographic [9,15,28], and histopathologic [6,9,11] evidence of osteoarthritic changes associated with PSB fracture, including PSB discoloration; PSB radiolucency and subchondral bone osteopenia, osteophytosis, and enlarged vascular channels; and PSB articular cartilage fibrillation and chondrone formation. Interestingly, a cadaveric magnetic resonance imaging (MRI) study revealed that horses sustaining PSB fracture were more likely to have evidence of orthopedic disease in the contralateral limb compared to non-fracture controls, including subchondral bone densification, disruption of the subchondral bone plate or articular cartilage of the palmar distal MC3 condyle, and intraarticular osteochondral fragmentation [15].

The objectives of this study were to investigate PSB density and mineral content using multiple modalities, including dual-energy X-ray absorptiometry (DXA), computed tomography (CT), Raman spectroscopy, and ash fraction measurements; and to investigate bone quality and MCPJ pathology in Thoroughbred racehorses with and without catastrophic PSB fracture. We hypothesized that horses sustaining PSB fracture would have greater bone mineral content and inferior bone quality as compared to controls and that cumulative high-speed furlong exercise would be predictive of bone volume fraction and bone mineral density measurements.

## 2. Materials and Methods

### 2.1. Study Population

Forelimbs from Thoroughbred (TB) racehorses that died or were euthanized on New York racetracks were ethically obtained and chronologically enrolled in the study from 2019 to 2020. The study population consisted of 29 animals, including cases that sustained unilateral PSB fracture (*n* = 14, Appendix A) and controls that died or were euthanized due to reasons unrelated to MCPJ fracture (*n* = 15, Appendix A). Horses with bilateral fractures and horses with concurrent fracture of the distal MC3 or proximal P1 were excluded from the study.

The fracture group consisted of 14 horses, including 8 females, 3 castrated males, and 3 intact males which ranged in age from 2 to 6 years with a median age of 3 years. The median age at death was 3 years and ranged from 2 to 6 years. The fracture group included horses that sustained a fracture of one or both PSBs in one forelimb, while bones in the contralateral limb remained intact (*n* = 14, 8 left forelimb fractures, 6 right forelimb fracture). The control group consisted of 15 horses, including 5 females, 7 castrated males, and 3 intact males which ranged in age from 2 to 6 years, with a median age of 4 years. The median age at death was 4 years and ranged from 2 to 6. The control group contained horses that were euthanized or died due to disease or injury unrelated to the fetlock joint including: cardiovascular collapse (*n* = 4), laminitis (*n* = 4), head trauma (*n* = 2), colic (*n* = 2), sudden death (*n* = 1), carpal fracture (*n* = 1), humeral fracture (*n* = 1).

For the epidemiologic portion of the study, the mean percentage of exercise-related fatalities for 2-year-old horses during the 6-year period leading up to the 2020 Saratoga race meet was compared with that for the 2020 Saratoga race meet.

### 2.2. Training and Racing Exercise History

For each fracture case and control, training and racing exercise history was curated using information provided by the New York State Gaming Commission (NYSGC). Data collected included total high-speed furlong workouts, defined as any session in which an animal runs one furlong in 14 s or faster (~14.3 m/s or faster). Total furlongs is a race training factor that can be altered in future training methods and has previously been used as a measure of exercise history for evaluating catastrophic PSB fracture [6,11]. Additionally, from the exercise history provided, career duration (weeks), total weeks of rest, total weeks of work, total career furlongs, total number of races, number of breaks greater than 8 weeks of no work, and career work to rest ratio were calculated. The exercise-related fatality rate at Saratoga racetrack during the 2020 race meet was sorted by age and compared to the mean exercise-related fatality rate sorted by age from the 2014–2019 Saratoga race meets, using the New York State Gaming Commission Equine Breakdown, Death, Injury and Incident Database.

### 2.3. Imaging Acquisition and Processing

All limbs were transected at the level of the mid-radius for imaging and stored at 4 °C for <24 h while awaiting imaging and dissection.

#### 2.3.1. Dual-Energy X-ray Absorptiometry

All limbs were analyzed using a Discovery A/216 DXA scanner (Hologic Canada ULC, Mississauga, ON, Canada). Images were acquired within a 42 cm sagittal plane that included the distal MC3, both PSBs, and the proximal first phalanx (P1). Images were analyzed using onboard software. Measurements included the area and radiographic bone density (g/cm^2^) of the distal MC3 condyle, the palmarodistal MC3 condyle, and the PSBs.

#### 2.3.2. Computed Tomography Acquisition

Images of all forelimbs (total *n* = 58: fracture *n* = 14, fracture contralateral *n* = 14, control *n* = 15, and control contralateral *n* = 15) were acquired with the limb’s long axis perpendicular to the scan axis. The image acquisition was centered on the PSBs and extended approximately 30 cm proximally (distal 1/3 of MC3) and distally (proximal 1/3–1/2 of the proximal phalanx, P1) from the metacarpophalangeal joint using 16-slice helical computed tomography (Toshiba/Canon Aquilion Large-Bore, Toshiba/Canon America Medical Systems, Tustin, CA, USA). Images were acquired using 135 kVp with a scan field of view 240 mm and a standard reconstruction kernel. Images were reconstructed using 0.5 mm slice thickness at a 0.3 mm slice interval, reconstruction field of view 240 mm, pixel dimensions ranging from 0.296 to 0.468 mm, and 512 × 512 display matrix. All images were reconstructed in a bone filter, displayed with window width 4500, window center 1100, and variable tube current (SURE Exposure^TM^).

#### 2.3.3. Computed Tomography Manual Analysis

Images were exported and analysed using Horos^TM^ (Nimble Co. LLC d/b/a Purview, Annapolis, MD, USA). The proximal first phalanx (P1), PSBs, and distal MC3 were assessed for signs of pathological changes, including osteophytes, osteochondral fractures or fragmentation, enthesiophytes, subchondral lysis, and subchondral cyst-like lesions. Distal MC3 measurements included areas of the medial and lateral condyles, depth, and area of sclerosis of the condyles, radiographic bone density as measured in Hounsfield Units (HU) of the complete condyle and of the sclerotic region, and standard deviation of the radiographic bone density of the distal condyle and sclerotic region in a sagittal plane centered on each condyle. All intact PSB bones were measured for maximum height, width, and depth. Area, radiographic bone density, and standard deviation of radiographic bone density were measured for all intact PSBs in the following planes: apical, mid-body, basilar, axial, mid-sagittal, abaxial, sub-condylar, medullar, and flexor (Appendix A). Fractured PSBs were not analyzed. Proximal P1 measurements included the depth of subchondral bone sclerosis measured in the frontal plane at the level of the sagittal grove and the medial and lateral thirds of the bone.

#### 2.3.4. Computed Tomography Radiomics Analysis

DICOM images were exported to 3D Slicer (v4.11) (Irvine, CA, USA) for subsequent segmentation and analysis (“3D Slicer”, n.d.). For each intact forelimb, four regions of interest (ROIs) were drawn on each forelimb scan: the visible MC3 within the image volume, the visible portion of proximal P1, and the medial and lateral PSBs. Because many CT scans did not include the entire MC3 or P1, two additional regions of interest were created for the MC3 and P1 bones that extended from the subchondral surface of the bone to approximately 5 cm either proximally or distally to the joint. Finally, a 15 mm spherical volume entirely contained within the PSBs was drawn, resulting in a total of eight ROIs analyzed. These ROIs are hereafter referred to as PSBs, MC3-Full, MC3-Partial, P1-Full, P1-Partial, and PSBs-15 mm.

Bone mineral density (BMD), bone volume fraction (BVF), and total mineral density (TMD) were extracted from the CT data with scripts written in Matlab, using Otsu’s method for thresholding image data [29]. The conversion of CT Hounsfield Units (HU) to K2HPO4 density was performed by sampling the voxel intensities over a 1 cm diameter by 2 cm long cylinder within the 2 cm diameter phantom plugs, and an average HU to K2HPO4 density (mg·cm^−3^) calibration curve was applied to the CT data. Tabularized data of features were exported and analyzed in JMP.

### 2.4. Dissection and Gross Examination

Following imaging, distal limbs from both fracture and control groups were dissected by a board-certified pathologist (SPM). Pathology descriptions included macroscopic observations of joint disease, including articular cartilage damage to the MC3, PSBs, and proximal P1 (none, mild, moderate, or severe) and the presence of osteochondral fragmentation. For fracture limbs, cartilage damage that was believed to be the result of trauma from the acute sesamoid fracture fragment was excluded to describe the health of the joint prior to fracture. Medial and lateral metacarpal condyles were assigned a grade for palmar osteochondral disease (POD), based on the degree of discoloration of the subchondral bone and disruption of overlying articular cartilage. Grades were assigned as follows: grade 0, no evidence of POD; grade 1, discoloration or subchondral bone without disruption of overlying articular cartilage; grade 2, discoloration of subchondral bone plus mild to moderate disruption of overlying articular cartilage; and grade 3, discoloration of subchondral bone plus collapse of overlying articular cartilage. As previously reported, the mean POD grade for each horse was calculated and reported as a categorical outcome ranging from 0 to 3 [30]. Following necropsy, PSBs were analyzed for fracture configuration and abaxial subchondral bone lesions characterized by discoloration or an articular cartilage or subchondral defect as recently described by Shaffer et al. [9]. The distal MC3, PSBs, and P1 were stored in saline-soaked gauze at −20 °C until further processing.

### 2.5. PSB Sectioning and Processing

The medial sesamoid of the contralateral limb of the fracture group (*n* = 8) and the medial sesamoid of a randomly assigned limb in the control group (*n* = 8) were selected for Raman spectroscopy. Samples were cut into five parasagittal sections using a low-speed saw (Buehler IsoMet, Buehler, Lake Bluff, IL, USA) and stored at −20 °C until further processing. A 2 to 4-mm-thick section just axial to the mid-sagittal section was selected for Raman spectroscopy to maximize the surface area. These sections were adhered to a metal sample holder using mounting wax (Allied High Tech Products Inc., Rancho Dominguez, CA, USA) and polished on an automated/semi-automated polishing system (MultiPrep^TM^, Allied High Tech Products, Inc., Rancho Dominguez, CA, USA), using decreasing particle sizes of diamond wafer paper (30-micron to 3-micron, Allied High Tech Products, Inc., Rancho Dominguez, CA, USA), followed by 0.3-micron alumina slurry (Allied High Tech Products, Inc., Rancho Dominguez, CA, USA). Samples were sonicated in isopropyl alcohol to remove abrasive particles, removed from the sample holder, wrapped in lens paper, and stored at −20 °C in saline-soaked gauze until the time of characterization with Raman spectroscopy.

### 2.6. Raman Spectroscopy

Raman spectra were acquired from the polished surfaces of mid-sagittal sections of the PSB. A confocal Raman microscope (WITec Alpha300R, WITech Instruments Corp., Ulm, Germany) with a 785 nm diode laser; 20×, 0.5 N.A. objective (Zeiss, Jena, Germany) and software that enables imaging guided by surface topography (TrueSurface Microscopy, WITech Instruments Corp., Ulm, Germany) was used to collect the spectra. The spot size was ~1900 nm. Nine rectangular regions in a 3 × 3 grid were identified on the sample surface (Figure 1A) and grouped for subsequent analysis both in the proximal-distal direction, to form the subchondral, medullary and flexor regions (Figure 1B) and in the dorsal-palmar direction to form the apical, midbody and basilar regions (Figure 1C). Three spectra were collected in each region in a line along the proximal-distal direction with a 1000 µm spacing between each collection point. Thus, a total of 27 spectra were collected from each sample. For each spectrum, the total acquisition time was 30 s.

Background fluorescence was subtracted from the Raman spectra using commercial software (WITec Project FIVE, WITech Instruments Corp., Ulm, Germany). All spectra were normalized to the ν_1_PO_4_ peak (960 cm^−1^). Raman spectra (Appendix A) were analyzed using custom code (Matlab, Mathworks, Natick, MA, USA) developed to calculate Raman compositional parameters **(**Appendix A) from integrated peak areas that characterize the bone mineral and matrix (Appendix A). The positions of several features that characterize the collagen matrix were determined through second derivative spectroscopy, including glycosaminoglycans (GAGs, derived from CH_3_ deformation [1365–1390 cm^−1^]) [31], extracellular matrix (derived from methylene side chains [CH_2_] deformation (1450 cm^−1^) [32,33], the mature trivalent enzymatic crosslink Pyridinoline (PYD, 1660 cm^−1^) [34,35], the immature divalent enzymatic crosslink dehydrodihydroxylysinonorleucine (de-DHLNL, 1690 cm^−1^) [35], and the AGE pentosidine (PEN, 1495 cm^−1^) [35,36,37]. After initial peak positions were determined from the second derivative spectra, Gaussian functions were fit to the Raman spectra using spectroscopy software (GRAMS/AI, Thermo Galactic, Waltham, MA, USA) (Appendix A). A non-linear least squares method was used to find the final peak position of each peak with a ±5 cm^−1^ window [35].

### 2.7. Ash Fraction

Three to four mm sagittal sections of the medial PSB from each case (*n* = 29, *n* = 14 fracture, and *n* = 15 control) were further sectioned into nine sub-regions: apical flexor, apical medullary, apical subchondral, mid-body flexor, mid-body medullary, mid-body subchondral, basilar flexor, basilar medullary, and basilar subchondral using a low-speed saw (IsoMet, Buehler, Lake Bluff, IL, USA). Each sub-region was individually placed into an Eppendorf microcentrifuge tube and stored at −70 °C prior to ash fraction measurements.

Initial weights for each of the nine PSB sections were recorded. New microcentrifuge tubes were obtained for each bone section and filled with acetone such that the volume was approximately 10× the volume of the bone. Bone sections were individually submerged, and microcentrifuge tubes were sealed with laboratory film (Parafilm). The tubes were placed within a fume hood in end-over-end rockers for 72 h, and the acetone in the microcentrifuge tubes was replaced every 24 h. Bone sections were subsequently removed from acetone and placed in individual porcelain crucibles (High-Form Porcelain Crucibles, Fisherbrand, Leicestershire, UK) using thumb forceps and dried in an oven (Horizontal Air Flow 1370 FM, VWR Scientific Products, Radnor, PA, USA) at 60 °C for 23 h, followed by measurement of dry weight (scale description and manufacturer here). The sections were ashed in a muffle furnace (Type 1300 Furnace, Barnstead Thermolyne Corp., Dubuque, TX, USA) at 600 °C for a period of approximately 18 h. After cooling in a desiccator (Pyrex, Corning, NY, USA) for approximately one hour, a final ash weight was recorded. Ash fraction measurements were calculated by dividing weight by ash weight. Percent mineral by weight was calculated by dividing ash weight by dry weight and multiplying by 100.

### 2.8. Statistical Analysis

To compare exercise variables between fracture and control groups, a paired t-test was performed for career duration, weeks of rest, weeks of work, total furlongs, total number of races, number of training breaks of ≥8 weeks, and the ratio of work weeks to rest weeks. To determine associations between age and exercise variables, Pearson’s correlation coefficients were calculated for age and these same exercise variables. To validate the repeatability of the measurements manually acquired from CT images by a boarded radiologist (IRP) and an equine veterinarian (KJN), an intra-class correlation coefficient (ICC) analysis was performed.

All nominal and ordinal data were assessed for normality. Nominal data were analyzed for differences between fracture and control groups using Fisher’s Exact tests, and ordinal data were analyzed using ChiSquare likelihood ratios. Continuous data obtained from DXA, CT, and ash fraction measurements were compared using linear mixed-effects models. The fixed effects consisted of group (fracture vs. control), sex, and high-speed furlongs (HSF). The random effects included horse, limb (left vs. right), and, when appropriate, laterality (medial vs. lateral). Radiomic features for the medial and lateral PSB were examined together when exploring differences in the case and control image sets, as previously described [9,11].

Linear mixed-effects models were also used to compare Raman parameters across the fracture and control groups and across the anatomic regions (Appendix A). Comparisons across anatomic regions were performed using three models: a model that included all nine anatomic regions (Figure 1A) and two additional, simplified models that included three anatomic regions grouped in the dorsal-palmar direction (creating subchondral/medullary/flexor regions) (Figure 1B) or in the proximal-distal direction (creating apical/midbody/basilar regions) (Figure 1C). Fixed effects included group (fracture vs. control), sex, total high-speed furlong workouts (HSF), region and the interaction between region and group; random effects included the sample to account for multiple spectra collected within each animal. A Tukey posthoc test was used for multiple comparisons of the estimated marginal means of the regions. The significance level was set to *p* = 0.05. Raman analyses were performed using R studio (R Studio, PBC, Boston, MA, USA).

Finally, a multivariate analysis with pairwise correlations was performed to compare between modalities and reported using Pearson’s correlation coefficients. This included a comparison between DXA radiographic bone density of the sesamoids in the lateral projection, CT-derived Hounsfield Units of the mid-sagittal PSB, radiomics derived bone mineral density of the mid-sagittal PSB, and ash fraction. To maintain independence of observations and assess correlations across modalities, mean values for combined left and right forelimbs were calculated as appropriate. All analyses were performed in JMP Pro 16 (Cary, NC, USA) unless stated otherwise.

## 3. Results

### 3.1. Study Population

Within the fracture group, 13 of 14 horses (92.9%) sustained fractures to both the medial and lateral proximal sesamoid bones, while one horse fractured only the medial PSB. Of the fractured sesamoids, 12 were fractured in the basilar region, 7 in the mid-body region, and 1 in the apical region.

For the 2014 through 2019 Saratoga race meets, 2-year-old horses represented a mean of 29% of the total number of exercise-related fatalities. During the 2020 Saratoga race meet, 2-year-old horses represented 82% of the total number of exercise-related fatalities. This represents a 183% increase in 2-year-old horse exercise-related fatalities in 2020.

### 3.2. Exercise Comparisons between Fracture and Control Groups

Exercise histories were available for 28 of the 29 horses in this study; training data were not available for one case except for cumulative total furlongs. Exercise history data were similar for fracture cases and controls (Table 1).

### 3.3. Association between Age and Exercise

Exercise variables, including career duration, total weeks of rest, total weeks of work, total furlongs, and the total number of races, were strongly positively correlated with horse age (Appendix A). As a result of the strong correlation between exercise variables and age, the total furlongs measure was selected as the variable to analyze in regression models [6,11].

### 3.4. Intra-Class Correlation Coefficients

The majority of intraclass correlation coefficients (ICC) for manual CT parameters identified moderate (0.5 to 0.75 ICC) to good (0.75 to 0.9 ICC) correlation between the two observers. For example, ICCs ranged from a high of 0.91 for frontal plane measurements of the PSB midbody area to a low of 0.35 for the mean area of sclerosis of the MC3 condyle believed to be due to the subjective nature of this area measurement. Complete ICC values are reported in Appendix A.

### 3.5. Effect of Group: Fracture versus Control

#### 3.5.1. Pathologic Features

Palmar osteochondral disease (POD), defined as a gross POD score of ≥1, was detected in 19 out of 22 (86.4%) horses in which necropsy POD score data were available. Of these 22 horses, 12 were fracture cases and 7 were controls. Grossly, no differences were noted for POD scores between fracture and control cases (*p* = 0.20). Discoloration of PSB articular cartilage was present in 7/14 fracture cases and 8/15 control cases (*p* = 1.0).

#### 3.5.2. Imaging Features

DXA bone mineral density measurements of the distal MC3 condyle, palmar distal region of MC3, and the PSBs did not identify any differences between fracture and control groups. However, the CT radiographic density of the basilar PSB region was more homogenous in fracture cases as compared to controls (fracture: 1211.7 ± 61.9, control: 1184.8 ± 70.7 HU; *p* = 0.049) which could indicate sclerosis. Dorsal cavitation of MC3 at the level of the proximal MCPJ was identified in 7 out of 15 control horses. Interestingly, this feature was not observed in any horses sustaining PSB fracture (*p* < 0.0001) (Figure 2A,E). No differences in osteophytes, enthesiophytes, or subchondral lysis of MC3, P1, or the PSBs were observed between fracture and control groups (Appendix A).

#### 3.5.3. Ash Fraction

Whole bone PSB percent mineral was greater than tuber coxae percent mineral (60.2 ± 1.3% versus 53.4 ± 3.9% (Appendix A). The PSB basilar medullary region had 0.48% less mineral in fracture cases as compared to controls (*p* = 0.02). Percent mineral measurements for the remaining eight sections did not differ between groups (*p* = 0.07 to 0.96). Regional comparison of all PSB mid-sagittal sections combined identified 2.44% great mineral in the medullary region compared to subchondral region (*p* < 0.001) and 4.01% greater mineral that the flexor region (*p* < 0.001). Additionally, the midbody region had 2.48% greater mineral compared to the apical region (*p* < 0.001) and 0.96% greater mineral than the basilar region (*p* < 0.05) (Appendix A). Tuber coxae samples were available for 26 of the 29 horses, including 13/14 fracture cases and 13/15 controls. The percent mineral of the tuber coxae did not differ between cases and controls (*p* = 0.94).

#### 3.5.4. Raman Spectroscopy

When the effect of group (fracture vs. control) was analyzed, all Raman spectroscopic parameters were similar across groups (Figure 3). The absence of differences between groups was consistently observed across all statistical models regardless of how the region was defined (9 sub-regions, 3 dorsal-palmar regions, or 3 proximal-distal regions), (*p* > 0.05), (Figure 4, Appendix A).

### 3.6. Effect of Exercise: Total High-Speed Furlongs

#### 3.6.1. Gross Pathologic Features

On gross examination, POD lesions were more severe in horses with more accrued total furlongs, with an increase in 1.0 grade per 100 furlongs (*p* < 0.0001) (Figure 2B,F). POD score data were not recorded in the necropsy reports of the remaining seven horses, including two fracture cases and eight controls.

#### 3.6.2. Imaging Features

Both DXA (Figure 5) and CT (Figure 6) identified increased BMD in horses with greater total furlongs. Distal MC3 condyle bone mineral density was greater by 0.1 g/cm^2^ per 100 total furlongs, and the palmar distal region of MC3 was greater by 0.2 g/cm^2^ per 100 total furlongs (*p* < 0.0001) (Figure 5A, Appendix A). Similarly, increased CT radiographic bone density was detected in the lateral condyle of MC3, with an increase of 2.0 HU per 100 total furlongs accrued (*p* < 0.0001). Radiomics CT analysis identified that distal MC3 condyle BMD was greater by 0.10 g/cm^2^ per 100 total furlongs (*p* = 0.005), and the PSB 15 mm bone mineral density was greater by 0.19 g/cm^2^ per 100 total furlongs (*p* = 0.03) (Figure 6A,B). CT identified pathological features of the MC3 condyle associated with more total furlongs, including condylar sclerosis (*p* < 0.0001) (Figure 2C,G) and subchondral lysis (*p* < 0.0001) (Figure 2D,H). The subchondral lysis score was greater by 0.9 per 100 total furlongs (*p* < 0.0001). A non-significant trend for greater PSB BMD with increased total furlongs on DXA was observed (*p* = 0.054) (Figure 5B). CT identified increased radiographic bone density in the subchondral region of the PSBs by 16.4 HU per 100 total furlongs accrued (*p* = 0.04).

#### 3.6.3. Ash Fraction

Subchondral mineral content (percent mineral) was greater in PSBs from horses with more accrued total furlongs (Appendix A). The midbody subchondral PSB region had 0.47% more mineral for each additional 100 total high-speed furlongs. No other regional differences in mineral content were detected as an effect of group or exercise, though there was a trend for increased mineral content in the mid-body region as an effect of total furlongs (*p* = 0.07, Appendix A). Tuber coxae percent mineral did not differ as a function of high-speed furlongs (*p* = 0.43).

#### 3.6.4. Raman Spectroscopy

The carbonate:phosphate ratio (*p* = 0.054, 9-subregion; *p* = 0.055, dorsal-palmar; *p* = 0.053, proximal-distal, Appendix A) demonstrated a trend towards significance as a function of total high-speed furlongs. Specifically, a one-unit increase in high-speed furlongs resulted in a 0.04% increase in carbonate:phosphate ratio. However, the effect of high-speed furlongs was not significant for any other Raman parameters (Appendix A).

### 3.7. Correlation Statistics

To assess correlation metrics of PSB bone mineral density across modalities, Pearson’s correlation coefficients were calculated. Ash fraction and DXA (r = 0.24), ash fraction and CT (r = 0.23), manual CT and DXA (r = 0.06), and radiomics CT and DXA (r = 0.14) data were only weakly correlated. Manual and radiomics CT values (r = 0.58) were highly correlated.

## 4. Discussion

Cumulative high-speed furlong exercise was strongly predictive of several bone mineral measurements and metacarpophalangeal joint pathologic changes across all modalities, including DXA, CT, Raman spectroscopy, and ash fraction measurements. BMD in the MC3 condyles and PSBs and percent mineral in the subchondral region of PSBs were greater in horses with more total furlongs. MCPJ pathology, including POD scores, MC3 condylar sclerosis, and MC3 subchondral lysis, was greater in horses with more total furlongs. Contrary to our hypothesis, there were no differences in BMD or Raman parameters between fracture and control groups. MC3 dorsal cavitation was detected in 7/15 control cases but was not detected in horses sustaining PSB fracture. Raman spectroscopy and ash fraction measurements revealed regional differences in PSB BMD and tissue composition, including greater mineral-to-matrix ratios in the subchondral and basilar PSB regions and greater carbonate-to-phosphate ratios in the flexor and apical PSB regions. Taken together, these data suggest that cumulative high-speed furlong exercise is a strong predictor of bone mineralization and joint pathology and that these effects predominate over differences between PSB fracture cases and controls.

Prior studies have demonstrated that MC3 [38,39,40] and PSB [5,6,8,11,41] morphologic features and fracture risk vary as a function of exercise [6,11]. For example, BMD was greater in the MC3 condyles in 2-year-old Thoroughbreds with more exercise [19], and bone volume fraction of the parasagittal groove of MC3 was greater in Thoroughbreds in training than those in rest [42]. Some differences in morphologic features, especially those related to bone mineral content, could be due to adaptive responses to exercise and may not be pathologic, although the degree that these responses are considered adaptive versus maladaptive is currently unknown. In both humans and horses, bone is highly responsive to exercise [43], with changes occurring as early as 8 weeks of training [44]. Noble et al. identified differences in volumetric bone mineral density and subchondral bone thickness of the proximal phalanx in racehorses with catastrophic proximal phalanx fractures as compared to both raced and unraced controls, including greater variance in bone mineral density, which may indicate a maladaptive response [45]. More advanced osteoarthritic changes and greater POD scores were associated with more accrued total furlongs in New York racehorses [6]. California racehorses sustaining PSB fracture spent more time in active racing and training, exercised for longer periods prior to their last rest period, exercised at higher intensities during the 12 months prior to their death, and accumulated greater distances in their career than non-fracture racehorse controls [5]. In the UK, PSB fractures were also more common in experienced racehorses with more starts than in those in their first season of racing [46].

In the current study, horses sustaining PSB fracture were approximately evenly divided between ≤3-year-old horses and >3-year-old horses, with 3 out of 14 horses (21%) sustaining catastrophic PSB fracture prior to their first race. In addition, there were no differences in measures of total exercise between fracture and control horses. While these findings suggest that exercise history may account for some PSB and MCPJ pathologic changes, they suggest that PSB fracture is not always an injury associated with excessive work. It is likely that some PSB fractures occur in horses that have not sufficiently modeled their PSBs prior to their first race. Of significance, in 2020, New York Thoroughbred racetracks were closed due to the COVID-19 pandemic from the first week in March until the first week in June. During this time, horses were not able to complete their regular training schedules in preparation for the 2020 Saratoga race meet (July 16–August 6). This disruption of high-speed exercise training was associated with a disproportionate increase in exercise-associated catastrophic fractures among 2-year-old Thoroughbred racehorses during the 2020 Saratoga race meet compared with that of previous years. This pandemic-associated anomaly in PSB fracture demographics could potentially account for the absence of differences in bone mineralization parameters identified in this cohort of NY racehorses as compared to prior NY studies [47].

Consistent with prior studies [19,42,48,49], BMD was greater in MC3 condyles from horses with more total furlongs as measured via DXA, CT, Raman spectroscopy, and ash fraction. In addition, quantitative CT bone mineral density measurements were greater in PSBs from horses with more total furlongs. Since high-speed exercise is known to be a strong stimulus for bone remodeling, it is not surprising that total high-speed furlongs was the predominant explanatory variable for MC3 and PSB BMD in our models. Interestingly, no differences were noted in tuber coxae BMD as a function of group or exercise, suggesting that the effects of high-speed exercise are less pronounced at this axial skeletal site. Within PSBs, increased BMD was most notable in the subchondral region as evaluated by CT and ash fraction. However, no differences were found in whole PSB BMD between fracture cases and controls. These results, combined with previous findings of focal decreased bone mineral density in the abaxial subchondral region of racehorses sustaining catastrophic PSB fracture [9], suggest that regional differences in bone and mineral properties may be better predictors of fracture risk than whole PSB changes.

Palmar osteochondral disease (POD) commonly affects Thoroughbred racehorses, with a reported prevalence as high as 80.4% [50]. In the present study, POD was detected on gross examination at necropsy in 19 out of 22 (86.4%) horses, with greater POD scores in horses with more total furlongs. POD scores have previously been associated with greater cumulative racing and training, suggesting that POD is a progressive disease that results from bone fatigue with repetitive loading [50]. POD lesions have previously been associated with microcracks in MC3 condyle subchondral bone [40,51] and PSB fracture [11] in racehorses. The only gross pathologic finding that differed between fracture and controls in the current study was dorsal MC3 cavitation, which was present in 7 of 15 controls and no fracture cases. Davis et al. previously described MC3 dorsal cavitation as a reliable radiographic marker of POD [27]; however, in the present study, dorsal cavitation did not correlate with POD scores nor total high-speed furlong exercise.

PSB mineral content and crystallinity did not differ as a function of total furlongs or group. Prior studies with higher-resolution micro-CT suggested that horses sustaining PSB fracture had increased bone volume fraction, leading to the hypothesis that tissue mineralization would also be increased in horses with PSB fracture [11]. Contrary to our hypothesis, bone tissue material properties were similar between fracture and control PSBs, consistent with a previous study that revealed similar BMD in whole bone and axial sub-regions of PSBs between fracture and control groups [12].

When examining the effect of region on material properties within all PSBs (fractures and controls combined), lower mineral content, lower advanced glycation end products (AGEs, including pentosidine and carboxymethylysine), and greater carbonate substitution were observed in the flexor region. In PSBs, the flexor region is subjected to high tensile forces, which may promote remodeling activity and decrease average tissue mineralization and AGEs concentration [52]. Lower tissue mineral and AGEs content are both indicators of younger tissue, while greater carbonate:phosphate ratios may be attributed to the loss of mineral or the increase of younger mineral with more substitution [53]. Lower tissue mineral and AGEs content also may contribute to greater toughness in flexor region, which would enable this region to be more durable under tension [54,55]. The subchondral region had the lowest carbonate:phosphate ratio, an intermediate mineral:matrix ratio, and trended toward the highest collagen maturity. These observations suggest that the subchondral region may have older tissue [55], consistent with reduced remodeling activity. Focal lesions [9] and microcracks [14,56] are more frequently observed in subchondral region and may predispose this site to fatigue fractures.

Correlation between modalities was considered weak, whereas CT manual and radiomics BMD measurements were highly correlated. CT provides better morphological and texture resolution than DXA or radiography, and access to standing equine CT units is increasing; however, challenges still exist, including image acquisition at high resolution and unifying image analysis across different platforms. Advances in quantitative analyses of large imaging data sets using radiomics [57] hold significant promise to mine the substantial amounts of data provided by volumetric imaging. Raman spectroscopy is not currently employed in vivo for clinical diagnostics, but several groups are investigating the possible future in vivo use of Raman for bone quality assessment in both humans and horses [24,58]. Ash fraction is the gold standard for measuring bone mineral content and is considered a superior marker of bone strength compared to radiographic measured bone volume fraction in humans [59]; however, ash fraction measurements are destructive and require an invasive bone biopsy. Here, ash fraction measurements served as a gold standard for comparison to DXA, CT, and Raman measurements of bone mineral content.

### Study Limitations Paragraph

Limitations of this study included the sample size of 29 horses (58 PSBs). While high-speed furlong exercise data were available for horses training and racing on NY racetracks, exercise history was unavailable for time periods where horses were not training at a NY racetrack facility. This study utilized clinical CT with an imaging resolution ranging from 0.296 mm to 0.468 mm pixels rather than micro-CT with resolutions ranging from 2.8 μm to 80 μm. However, clinical CT has practical application in live horses, especially with the increasing availability of standing CT in practice. Of the total 29 samples, Raman spectroscopy was only performed in 16 horses, and both Raman spectroscopy and ash fraction measurements were performed on a single parasagittal section of the proximal sesamoid bone rather than the whole bone. Despite these limitations, our analyses combined complementary analyses across multiple levels of bone structural hierarchy to clinical specimens. The novel application of Raman spectroscopy to study bone tissue from horses with PSB fractures provided information unobtainable from existing clinical imaging modalities, including mineral properties such as carbonate substitution and matrix properties such as AGEs and glycosaminoglycan (GAGs) content which impact bone mechanical properties and are associated with fracture incidence in humans [36,37,60].

## 5. Conclusions

Cumulative high-speed furlong exercise was strongly predictive of several bone mineral measurements and metacarpophalangeal joint pathologic changes across multiple modalities. However, few differences were detected between fracture and control groups, suggesting that whole bone mineral properties or metacarpophalangeal joint pathology are not alone sufficient for identifying horses at risk of PSB fracture. Recent work suggests that focal or regional changes in bone mineral content, such as the presence of osteopenic subchondral PSB lesions, may be associated with fracture risk [9]. Unfortunately, current equine standing imaging modalities may lack the resolution required to identify changes at this length scale. Functional imaging assessments, such as positron emission tomography (PET), may hold more potential for identifying these small defects through metabolic changes which often precede structural changes observed with CT [61]. PET scanning has been shown to detect radiopharmaceutical uptake in the dorsoaxial articular surface and abaxial border of PSBs and palmar subchondral bone of the lateral MC3 condyles in racehorses which was undetectable by CT or MRI [61]. Nonetheless, PET scanning in horses is recommended to be used in conjunction with CT [61], and identifying parameters correlated with PSB fracture using both modalities could improve the accuracy of identification of horses at risk for fracture.

## Figures and Tables

**Figure 1 animals-13-00827-f001:**
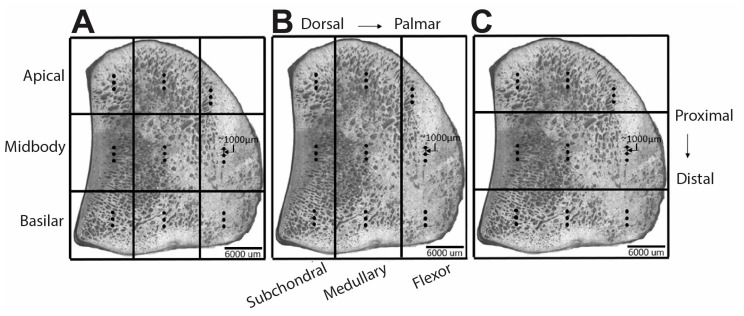
Representative optical micrographs of the mid-sagittal plane of the proximal sesamoid bone showing the 9 regions into which each specimen was subdivided for regional analysis by Raman spectroscopy. Three spectra (represented by solid circles) were collected within each region. (**A**) nine rectangular regions with a 3 × 3 grid layout; (**B**) three dorsal-palmar regions; (**C**) three proximal-distal regions.

**Figure 2 animals-13-00827-f002:**
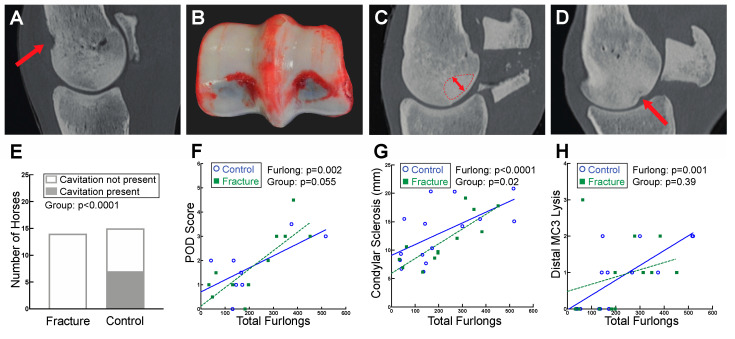
Evidence of palmar osteochondral disease (POD) was found on both gross pathology and CT analysis of the distal MC3. (**A**) Sagittal plane CT image of a control case with dorsal cavitation (indicated by the red arrow) of MC3 at the level of the proximal MCPJ. (**B**) Gross photograph of a fracture case with grade 3 POD of MC3. (**C**) Sagittal plane CT image demonstrating the region analyzed for condylar sclerosis measurements (dashed line with arrow indicating depth measured). (**D**) Sagittal plane CT image depicting distal MC3 subchondral lysis (indicated by red arrow). (**E**) Dorsal cavitation of MC3 at the level of the proximal MCPJ was identified in 7 out of 15 control horses but was not present in any of the fracture horses. (**F**) Gross POD scores increased with greater total furlongs. (**G**) The area of condylar sclerosis was larger in horses with greater total furlongs. (**H**) Distal MC3 lysis scores were higher in horses with greater total furlongs.

**Figure 3 animals-13-00827-f003:**
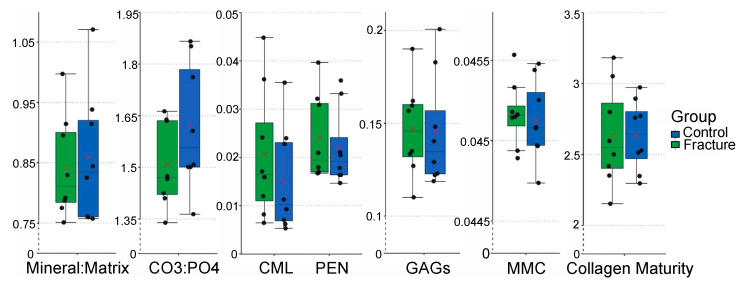
Box plots with jittered points of comparing the Raman spectroscopic outcomes of the fracture and control groups (*n* = 8 samples/group) using the nine sub-region model. In each box, red cross marks indicate the estimated mean value, and horizontal lines indicate the median value of each parameter in each group. All Raman spectroscopic outcomes were similar across groups (Abbreviations: CO_3_:PO_4_—Carbonate:Phosphate Ratio; CML—Carboxymethyl-lysine; PEN—Pentosidine; GAGs—Glycosaminoglycans; MMC—Mineral maturity/crystallinity).

**Figure 4 animals-13-00827-f004:**
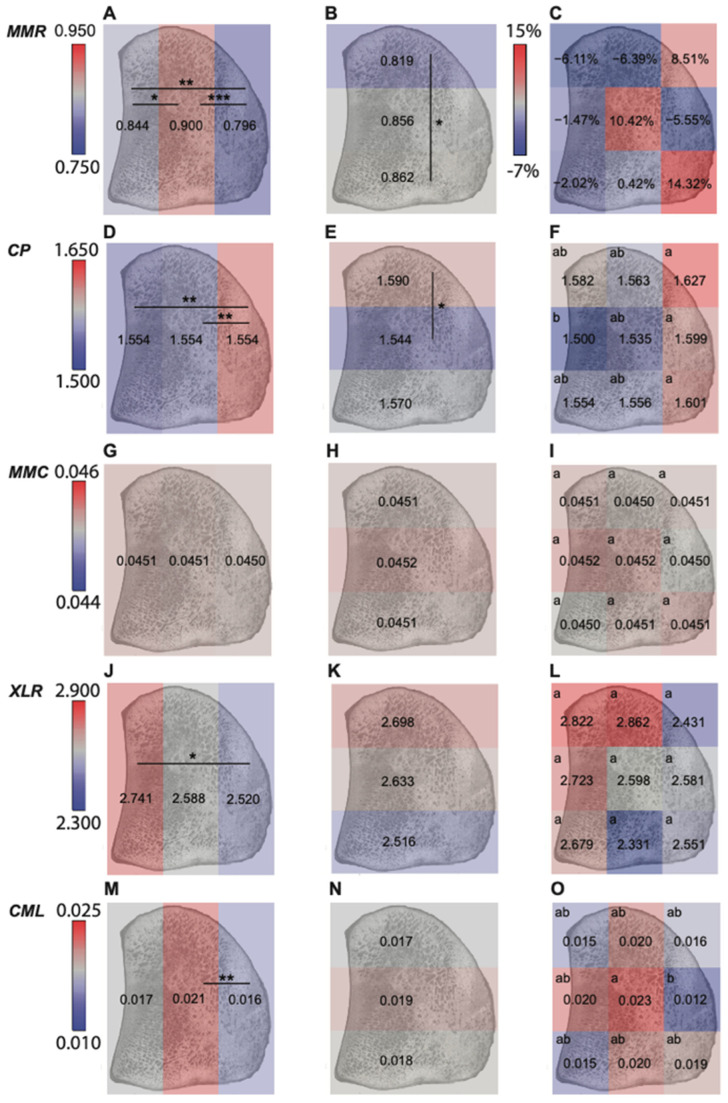
Regional comparison results within a mid-sagittal section of the proximal sesamoid bone of (**A**) mineral:matrix ratio across 3 dorsal-palmar regions; (**B**) mineral:matrix ratio across 3 proximal-distal regions; (**C**) mineral:matrix ratio across 9 anatomic sub-regions; (**D**) carbonate: phosphate ratio across 3 dorsal-palmar regions; (**E**) carbonate: phosphate ratio across 3 proximal-distal regions; (**F**) carbonate: phosphate ratio across 9 anatomic sub-regions; (**G**) mineral maturity/crystallinity across 3 dorsal-palmar regions; (**H**) mineral maturity/crystallinity across 3 proximal-distal regions; (**I**) mineral maturity/crystallinity across 9 anatomic sub-regions; (**J**) XLR across 3 dorsal-palmar regions; (**K**) XLR across 3 proximal-distal regions; (**L**) XLR across 9 anatomic sub-regions; (**M**) CML across 3 dorsal-palmar regions; (**N**) CML across 3 proximal-distal regions; (**O**) CML across 9 anatomic sub-regions. Results in **C** were shown as percent difference (fracture vs. control) to account for the significant interaction of group with region, others were shown as grouped means from both fracture and control group (Appendix A) (Abbreviations: MMR—mineral:matrix ratio; CP—carbonate:phosphate ratio; MMC—mineral maturity/crystallinity; XLR—collagen maturity; CML—Carboxymethyl-lysine) (***: *p* < 0.001; **: *p* < 0.01; *: *p* < 0.05). In the right column, there are significant differences across regions with no same letter.

**Figure 5 animals-13-00827-f005:**
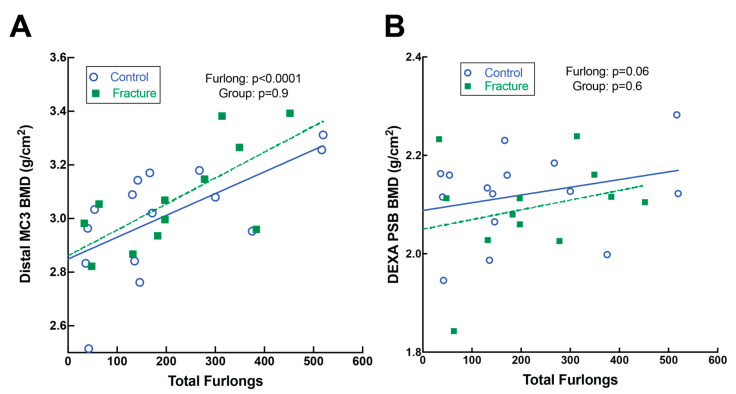
Dual-energy X-ray absorptiometry (DXA) analysis of the distal metacarpus (MC3) and PSBs. (**A**) Distal MC3 bone mineral density (BMD) did not differ between fracture and control groups but was greater in horses with more total furlongs. (**B**) Proximal sesamoid bone (PSB) BMD did not differ between fracture and control groups nor in horses with more total furlongs, though there was a non-significant trend for greater BMD with more total furlongs.

**Figure 6 animals-13-00827-f006:**
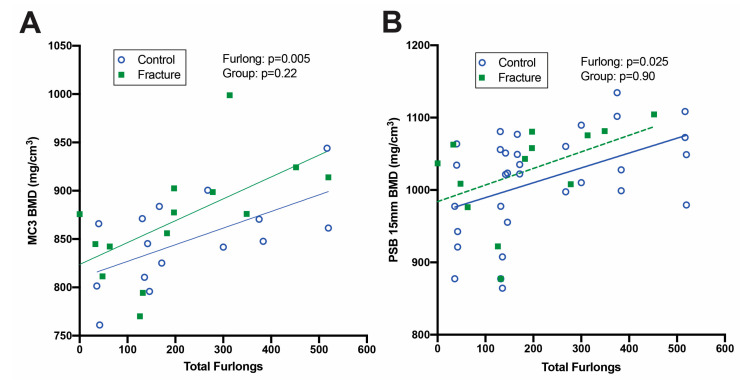
Linear regression analysis of CT images analyzed via radiomics. Bone mineral density (BMD) increased with total furlongs in both the distal third metacarpal bone (MC3) (**A**) and proximal sesamoid bones (PSB) (**B**) but did not differ between fracture cases and controls.

**Table 1 animals-13-00827-t001:** Exercise variables acquired for fracture and control cases as provided by the New York State Gaming Commission, reported as medians followed by ranges in brackets.

Exercise History Variable	Fracture	Control	*p*-Value
Age at first start	2 years (2 years)	2 years (2 to 3 years)	0.16
Career duration	64 (0 to 193) weeks	69 (12 to 212) weeks	0.92
Race starts	7.5 (0 to 28)	5 (0 to 40)	0.95
Cumulative total furlongs	190 (0 to 384)	146 (36 to 520)	0.91
Career work weeks	40.5 (5 to 96)	32 (8 to 95)	0.76
Career rest weeks	34 (2 to 112)	35 (4 to 123)	0.95
Career work:rest ratio	1.25 (0.59 to 2.35)	1.07 (0.54 to 2.45)	0.30

## Data Availability

All relevant data are published within the manuscript and Appendix A.

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
