# Peer review of "Metacarpophalangeal Joint Pathology and Bone Mineral Density Increase with Exercise but Not with Incidence of Proximal Sesamoid Bone Fracture in Thoroughbred Racehorses"

_animals, 2023, doi:10.3390/ani13050827_

Round 1
Reviewer 1 Report
The present study was well designed with clearly stated hypotheses. Potentially the research could provide insight into predisposing factors that lead to proximal sesamoid bone fractures and evaluate imaging methods that may be useful in vivo. The methodology and statistical analysis methods were well described and concisely written. The collected data was extensive and the tables and figures were easily interpreted. The authors reviewed the data objectively and came to valid conclusions. Overall this paper was very well written.

Author Response
Dear Reviewer,
We thank the reviewer for your time and attention to this manuscript and are thankful for the opportunity to address your concerns. Please find our detailed responses to these concerns and others listed below.
Response to specific comments:
Line 86 Where is reference #29 – References have been corrected so in numerical order.
Line 125 Could authors just state, awaiting imaging and dissection? – Thank you, this has been updated to reflect this suggestion.
Line 128 Add space between 42 and cm, or remove spaces in subsequent text. – Corrected
Line 146 Please add location (city, state, country) of Horos company – Requested information has been added.
Line 147 Change charges to changes – Corrected.
Line 161 Include 3D Slicer headquarters location (Irvine, CA) – Requested information has been added.
Line 162 Define ROI – Requested information has been added.
Lines 309-334 Much of the information in 3.1 should be included in section 2.1 (methods) rather than in Results (3.1). Certainly result information on specific fractures etc. could be in 3.1, but authors should work to reduceredundancy. Data on % of 2 year olds sustaining fractures by year / race meet is an important ‘result’(example). – Thank you, this information has been moved to the methods section of the manuscript and supplemental materials re-numbered to account for this change.
Reviewer 2 Report
The manuscript is meticulously and well described. It is interesting to read such a well-written article. My minor comments concern Fig.2, E-H and Fig. 3-they are illegible in the current version, they should be improved. Conclusions should come from one's own research, so I would ask that in this section you do not cite literature but focus on drawing (there is much to draw from) conclusions from your own analysis.
Regards
Author Response
Dear Reviewer,
We thank the reviewer for your time and attention to this manuscript and are thankful for the opportunity to address your concerns. Figures 2 and 3 have been amended to include large font sizes to make them easier to read. Additionally, the cited literature related to total furlongs has been removed from the results and moved to the methods section of the paper.
Reviewer 3 Report
Congratulations for this study and manuscript. The manuscript is well written, and this study is complex and very fascinating. They are very important in the equine medicine area
Study Population
The description of the groups is not clear.
All horses were euthanized because of PSB fracture?
What horse’s age?
How many races had they already run?
Were they in uninterrupted and intense training program?
Were control horses evaluated for bone/joint disease of MCPJ? (even if it was not the cause of euthanasia/death)?
What are the exclusion criteria?
*Part of this information are (only) in the “Results”. Should be better defined in the methodology
Training and racing exercise history
The compilation of training data is clarified, but there is a lack of information about the inclusion and exclusion criteria of experimental group.
The inclusion of possibility of joint adaptive process is missing, especially these young horses.
What could be disease and what is just joint adaptation to exercise/training? Did all these horses have clinical presentation of joint disease/ lameness/pain?
Bone/joint image alterations may not mean disease, but adaptation to the imposition of intense exercises program.
Therefore, inclusion and exclusion criteria, as well as training and age details are very important.
Perhaps, the comparison could have been made with untrained joints, or with different training intensities (group grading)
Author Response
Dear Reviewer,
We thank the reviewer for your time and attention to this manuscript and are thankful for the opportunity to address your concerns. Please find our detailed responses to these concerns and others listed below.
Study Population
The description of the groups is not clear. – Descriptions of both fracture and control groups have been moved to the methods section of the paper with additions to clarify inclusion and exclusion criteria, in addition to the demographic information contained in Tables 1A and 1B and Supplemental 4. If anything is still not clear, please let us know what additional information we can provide.
All horses were euthanized because of PSB fracture? – Yes; all horses in the fracture group were euthanized because of PSB fracture. Additional exclusion criteria have been added to lines 164-165 for clarification.
What horse’s age? – Horse age data has been moved to the methods section in line 168 and 173 as suggested by both Reviewers 1 and 3 and is included in Supplemental 4.
How many races had they already run? – Race data for each individual horse is included in Supplemental 1.
Were they in uninterrupted and intense training program? All horses were actively engaged in racing or race training exercise with the intent to race, at least until the proximate cause of death in a subset of the control horses. Additional data on exercise “interruptions” was add to Table 1, which includes information on work weeks, rest weeks, and work:rest week ratios. This work, rest, and work:rest week ratio data provides information on interruptions at the resolution that we are able to access based on the NYSGC exercise history database.
Were control horses evaluated for bone/joint disease of MCPJ? (even if it was not the cause of euthanasia/death)? – Yes, all horses underwent full necropsy, including evaluation of the MCPJ by the same boarded veterinary pathologist. This has been clarified in line 349.
What are the exclusion criteria? – Horses with bilateral fractures and horses with concurrent fracture of the distal MC3 or proximal P1 were excluded from fracture group. Horses with any fractures or abnormality related to the MCPJ were excluded from the control group. The exclusion criteria have been updated in lines 164-165.
*Part of this information are (only) in the “Results”. Should be better defined in the methodology. – As suggested, much of this information has been moved to the methods section in lines 160-180, with only the results of specific fracture configurations remaining in the results.
Training and racing exercise history
The compilation of training data is clarified, but there is a lack of information about the inclusion and exclusion criteria of experimental group. – Thank you; exclusion criteria have been expanded in lines 164-165 for clarification.
The inclusion of possibility of joint adaptive process is missing, especially these young horses. – Thank you for bringing up this important point. We have added a discussion of adaptive vs. maladaptive changes and the challenges in distinguishing between the two, especially in young horses, in lines 756-764 in the discussion section of the manuscript.
What could be disease and what is just joint adaptation to exercise/training? Did all these horses have clinical presentation of joint disease/ lameness/pain? – This is a great question and a question that is challenging to address as horses in both fracture and control groups may possess a combination of adaptive and maladaptive changes. In the horses that sustain catastrophic PSB fracture (especially those 2-year-olds and unraced horses), we would expect that some changes are certainly maladaptive. One of the goals of the study was to try to identify maladaptive change in horses sustaining PSB fracture; however, our results suggest that it may be more challenging than we thought since there were few differences between fractures and controls using these modalities. Most of the observed differences were a function of high-speed exercise and not fracture.
With respect to the clinical presentation of these horses, we unfortunately cannot answer this question due to the nature of the cadaver study design. As horses presented post euthanasia or death, we could not evaluate clinical presentation and we do not have access to medical records for these horses prior to the date at which the cadavers become the property of the New York State Gaming Commission (NYSGC). The only pre-euthanasia/pre-death data that we have access to is the training and exercise data collated by the NYSGC.
Bone/joint image alterations may not mean disease, but adaptation to the imposition of intense exercises program.
We absolutely agree that this is an important point and a point which was under-emphasized in the previous discussion section. We have added the following short paragraph to the discussion to address this important point in lines 755-761:
Some differences in morphologic features, especially those related to bone mineral content, could be due to adaptive responses to exercise and may not be pathologic, although the degree that these responses are considered adaptive versus maladaptive is currently unknown. In both humans and horses, bone is highly responsive to exercise [37], with changes occurring as early as 8 weeks of training [38]. Noble et al. identified differences in volumetric bone mineral density and subchondral bone thickness of the proximal phalanx in racehorses with catastrophic proximal phalanx fractures as compared to both raced and unraced controls, including greater variance in bone mineral density, which may indicate a maladaptive response [39].
Therefore, inclusion and exclusion criteria, as well as training and age details are very important.”
Yes, we very much agree on this point and have updated the methods section to include age details and inclusion/exclusion criteria in addition the detailed demographic information provided in Tables 1A and B and Supp. 1. In addition, we have added exercise history (training) data to table 1 to include career work weeks and career rest weeks (Line 501).
Perhaps, the comparison could have been made with untrained joints, or with different training intensities (group grading). – Yes, this is a limitation of this study as any of the included horses could, in theory, have fractured a PSB had they continued to train or race. We agree that a comparison with untrained joints or different training intensities would be exceptionally useful and, in fact, we have attempted to obtain these samples over the time period of this study and through the current date. Unfortunately, we have not been successful obtaining these samples from Thoroughbreds. Comparisons to other young, untrained, light breed horses is possible, but introduces substantial variability due to breed/genetic differences, and information on training/exercise programs is unavailable for these populations that we have access to. For this reason, we agree that this is an ideal goal, but not currently feasible for our group.